# Temporal Microbial Dynamics in Feces Discriminate by Nutrition, Fecal Color, Consistency and Sample Type in Suckling and Newly Weaned Piglets

**DOI:** 10.3390/ani13142251

**Published:** 2023-07-09

**Authors:** Barbara U. Metzler-Zebeli, Frederike Lerch, Fitra Yosi, Julia Vötterl, Juliane Ehmig, Simone Koger, Doris Verhovsek

**Affiliations:** 1Unit Nutritional Physiology, Department of Biomedical Sciences, University of Veterinary Medicine Vienna, 1210 Vienna, Austria; frederike.lerch@vetmeduni.ac.at (F.L.); fitra.yosi@vetmeduni.ac.at (F.Y.); julia.voetterl@vetmeduni.ac.at (J.V.); juliane.ehmig@vetmeduni.ac.at (J.E.); 2Christian-Doppler Laboratory for Innovative Gut Health Concepts of Livestock, University of Veterinary Medicine Vienna, 1210 Vienna, Austria; simone.koger@vetmeduni.ac.at; 3Department of Animal Science, Faculty of Agriculture, University of Sriwijaya, Palembang 30662, Indonesia; 4Institute of Animal Nutrition and Functional Plant Compounds, Department for Farm Animals and Veterinary Public Health, University of Veterinary Medicine Vienna, 1210 Vienna, Austria; 5University Clinic for Swine, Department for Farm Animals and Veterinary Public Health, University of Veterinary Medicine Vienna, 1210 Vienna, Austria; doris.verhovsek@vetmeduni.ac.at

**Keywords:** microbial colonization, fecal color, fecal consistency, fecal swab, sow milk, creep feed, piglet

## Abstract

**Simple Summary:**

The collection of feces has several advantages when studying the development of the gut microbiome in piglets as feces are easy to collect and can be collected from the same animal several times. Sow milk is the major food for the piglet and microbes during the suckling phase. The introduction of creep feed during the suckling phase may modify age-related microbial development. Moreover, feces from healthy piglets can differ in consistency and color. Moreover, it can be difficult to always collect feces from very young piglets, in which cases rectal swab samples are often collected. These factors may be behind the variation in microbiome composition and total abundances among animals. We followed the developing microbiome in feces from day 2 to 34 of life in piglets that drank only sow milk or had additional access to creep feed during the suckling phase. Weaning took place on day 28. Results supported that age and nutrition during the suckling phase were major factors influencing total and relative microbial abundances. However, results also indicated that color, consistency and sample type should be considered as additional factors when studying the fecal microbiome in suckling and newly weaned piglets.

**Abstract:**

Feces enable frequent samplings for the same animal, which is valuable in studies investigating the development of the gut microbiome in piglets. Creep feed should prepare the piglet’s gut for the postweaning period and shape the microbiome accordingly. Little is known about the variation that is caused by differences in fecal color and consistency and different sample types (feces versus swab samples). Therefore, this study evaluated the age-related alterations in the microbiome composition (16S rRNA gene) in feces of suckling and newly weaned piglets in the context of nutrition and fecal consistency, color and sample type from day 2 to 34 of life. Feces from 40 healthy piglets (2 each from 20 litters) were collected on days 2, 6, 13, 20, 27, 30 and 34. Weaning occurred on day 28. Half of the litters only drank sow milk during the suckling phase, whereas the other half had access to creep feed from day 10. Creep feeding during the suckling phase influenced the age-related total bacterial and archaeal abundances but had less of an influence on the relative bacterial composition. Results further showed different taxonomic compositions in feces of different consistency, color and sample type, emphasizing the need to consider these characteristics in comprehensive microbiome studies.

## 1. Introduction

Environmental and maternal bacteria from the vagina, nipple surface and milk colonize the neonatal porcine gut immediately after birth [1]. Concurrently, several complex internal and external factors interfere with early colonization by modulating ecological successions [1]. Early-life nutrition, antibiotics and (maternal) stress are major influencing factors for gut microbiome development [2,3]. An abnormal development may compromise the establishment of a stable and diverse gut microbiota, immune competence and growth performance and may increase disease susceptibility [3,4]. Any disruption in the early gut colonization can lead to overgrowth of pathobionts and induction of proinflammatory status [5,6]. Dietary modulation has been implemented to favorably shape the early colonization of the piglet’s gut from the pre- to postnatal phase in order to prevent dysbiosis after weaning [7,8]. Creep feeding of piglets during the suckling phase aims to prepare the digestive and secretory functions as well as the gut microbiome for the consumption of plant-based feed before weaning. The slowly evolving microbial changes with increasing creep feed intake should prevent the loss of immune tolerance towards the novel composition of the gut microbiome which often occurs after weaning. This situation is characterized by gut inflammation and compromised digestive and gut barrier functions [9]. However, the creep feed intake can be very low under practical farm conditions; hence, benefits may not always be observed [10].

To properly characterize the early development of the gut microbiome and its priming effect on the host, frequent sampling is required [11,12]. Feces are easy to collect, but most of all they enable the longitudinal time-series study of the gut colonization pattern in the same animal [3,13]. Although the fecal microbiome has no significance for the upper gut, certain similarities between the microbial composition found in the distal parts of the large intestine and feces exist [14]. While it is possible to collect intestinal samples in studies with pigs from euthanized animals, different animals are sampled, which increases the inter-animal variability, and hence, more animals are needed [11,15]. Therefore, the determination of the fecal microbiome is acceptable for repeated sampling schemes [11]. Literature consistently shows that *Enterobacteriaceae*, *Fusobacteriaceae* and *Clostridiaceae* are early colonizers detectable in feces, which are followed by *Bacteroidaceae*, *Lachnospiraceae* and *Lactobacillaceae* [7,16]. The early microbial composition and microbe–microbe interactions are shaped by species-specific milk glycans (lactose and milk oligosaccharides) that serve as substrates [12]. They increase the abundance of milk glycan-fermenting taxa, such as *Lactobacillaceae* and *Bacteroidaceae* [12,17]. After weaning, porcine milk glycans are abruptly removed from the diet, leading to a drop in milk glycan-fermenting taxa [12] and an increase in plant-starch- and plant-fiber-degrading taxa, e.g., *Prevotellaceae* [16,17]. In creep-fed piglets, it can be assumed that microbial taxa capable of utilizing plant starches and fibers can be detected in feces before weaning, corresponding to the actual creep feed intake of the piglet.

When collecting feces, several aspects are important to consider. While contamination, sample handling, freezing and thawing conditions can be controlled to a great extent in pig experiments, feces may not be always collectable from very young piglets, rendering it necessary to utilize rectal swabs [11]. Rectal (mucosa) swabs and feces represent different microbial communities; therefore, the comparability between the two types of samples may be low [11]. Contrasting this assumption, Choudhury et al. [11] proposed that rectal swabs are a suitable alternative sample type in young piglets. They based their conclusion on the fact that both types of samples provided relatively similar microbiome profiles with greater inter-individual variation than sample-type variation. Nevertheless, we assume that the (mucosal) swab samples comprise more glycoprotein-degrading taxa, such as *Clostridiaceae*, *Campylobacteriaceae* and *Veillonellaceae*, than fecal samples.

Other characteristics of feces that receive very little attention in microbiome studies but may be helpful in explaining inter- and intra-litter variation are fecal consistency and color. Fecal consistency often varies within and between litters, ranging from small pellets (constipation) to soft feces. Differences in consistency may be either related to the microbial composition or metabolites that act osmotically [8]. The determination of the dry matter content in feces can help analytically correct for different water contents, but the physiological effect on the host is associated with the wet digesta before defecation. For instance, very dry feces are usually retained longer in the gut, with longer mucosal exposure to microbes and their metabolites. Moreover, from an analytical point of view, the amount of feces can be limiting in very young piglets and may not enable an additional dry matter analysis. Only few studies have been conducted that investigated early microbial colonization patterns in piglets in relation to fecal consistency, often making the main difference between diarrheic and non-diarrheic piglets [18]. Diarrhea has various causes, all of which are reflected in imbalances of the commensal gut microbiome [18]. A similar description is missing for feces of different consistency in non-diarrheic piglets. We assume that in non-diarrhetic piglets, dry and soft feces also discriminate between their most prominent taxa. Feces can also differ in color, changing from yellowish to a dark brown, which may be related to bile acid turnover and/or microbial activity. However, there is limited information available on to which degree the change in color is age-related and a sign of digestive maturation or a sign of a change in the diet (e.g., introduction of creep feed). To investigate this, we hypothesized that darker feces would comprise more plant-degrading taxa, e.g., *Prevotellaceae*, compared to yellowish-colored feces.

In the present study, we evaluated the age-related alterations in the bacterial microbiome composition in feces of suckling and newly weaned piglets in the context of nutrition (sow milk only versus additional creep feeding), fecal consistency (very soft, soft and balls), color (yellow, brown and grey) and type of sample (swab versus feces and meconium) during the suckling and newly weaned period. We further assessed total bacterial, archaeal, protozoal and fungal abundances as a measure of temporal microbial dynamics in feces using quantitative PCR. This was completed by partial least squares discriminant analysis (PLS-DA) to identify genera that discriminated the fecal microbiome of piglets according to their nutrition during the suckling phase, fecal consistency and color, as well as the sample type from day 2 of life to one week postweaning. We chose to conduct this study under production conditions as experimental conditions can differ from the situation in practice [7].

## 2. Materials and Methods

### 2.1. Animals, Housing and Feeding

The animal experiment was conducted under production conditions at the pig facility of the research and teaching farm “VetFarm” of the University of Veterinary Medicine (Vienna, Austria). The experiment consisted of two replicate batches, with ten sows (Large White) per replicate batch. Sows from parities 1 to 6 were included. The sow management including housing and feeding was in accordance with the regular protocol at the pig facility, and a detailed description of the conditions can be found in the work of Metzler-Zebeli et al. [19]. Briefly, five days before the calculated farrowing date, sows were moved to the farrowing room where they were housed individually in Befree pens (BeFree, Schauer, Agrotonic, Prambachkirchen, Austria; 2.3 m × 2.6 m) until weaning. Each of these pens comprised a feeder, a hayrack, drinker and a piglet nest with heated flooring. Sows (Large White) were not constrained for farrowing, and all gave birth within 48 h. Low-birth-weight piglets were cross-fostered on day 1 of life to sows that were not included in the present study in order to standardize the litter size to 13 piglets. Male piglets were castrated on day 11 of life after sedation (Stresnil 40 mg/mL, 0.025 mL/kg body weight, Elanco Tiergesundheit AG, Basel, Switzerland, and Narketan 100 mg/mL, 0.1 mL/kg body weight, Vetoquinol Österreich GmbH, Vienna, Austria). On day 28 of life, piglets were transferred to the weaner pig room. Piglets were pen-housed in groups of a maximum of 20 piglets (pen size 3.3 m × 4.6 m), and piglets from two to three litters of the same feeding group from the suckling phase were housed together. Pens were equipped with a piglet nest, nipple and bowl drinkers and one round feeder. Straw was provided as bedding material. Throughout the experiment, sows and piglets had free access to water.

All diets were commercial complete feeds that met the current recommendations for nutrient requirements [20] and were the standard diets fed at the pig facility. From the week before farrowing and throughout lactation, sows were fed commercial cereal–soybean meal-based diets (Appendix A). The 20 litters were divided into two dietary groups which were balanced for the parity of the sow. In both piglet feeding groups, the piglets were reared on the sow until weaning (day 28 of life). Ten litters (*n* = 5/replicate batch) only drank sow milk, whereas the other ten litters (*n* = 5/replicate batch) were offered creep feed from day 10 of life. The creep feeding followed the regular feeding protocol for suckling and weaned piglets at the pig facility. For two weeks, the piglets in the creep-fed group were offered a commercial bovine whey powder-based milk replacer ad libitum (Appendix A). The milk replacer consisted of a powder that was mixed 5:1 (*w*/*v*; 200 g/L) with warm water (40 °C) according to the manufacturer’s instructions to produce a thin liquid that was manually served and offered in piglet feed troughs at least twice daily (800 and 1500 h). On days 24 and 25 of life, the prestarter diet was gradually mixed under the creep feed (Appendix A) to 100% and fed to the piglets from day 26 to day 35 of life. Piglets in the control group were offered the prestarter diet only after weaning from day 28 to 35 of life. Piglets in the respective feeding groups had ad libitum access to the creep feed and/or prestarter diet.

### 2.2. Body Weight Measurements and Collection of Fecal Samples

The experiment was conducted with a cohort of 40 piglets from the 20 litters. Two piglets per litter, balanced by sex, were selected based on the average birth weight on a litter basis at day 2 of life. The body weight (BW) of all piglets was measured on days 1 (birth weight), 2, 6, 13, 20, 27, 30 and 34 of life. Feces were collected on days 2, 6, 13, 20, 27, 30 and 34 of life. None of the selected piglets received medical treatment during the suckling and early postweaning phase. Also, none of the piglets developed liquid diarrhea. The fecal samples were obtained after rectal stimulation by inserting the tip of a sterile cotton swab into the rectum and gently turning the swab against the internal anal sphincter. Defecated feces were placed into a 1.5 mL Eppendorf tube, and color and consistency were recorded. In case no defecation occurred during 1 or 2 min, the swab was withdrawn, the stick was broken and only the cotton swab was placed into a 1.5 mL Eppendorf tube. The type of fecal sample (swab or feces) was recorded. Fecal colors comprised dark brown (meconium), yellow, grey and brown. Fecal consistency was scored during samplings (balls, soft (normally shaped) and very soft (but still shaped)). The tubes were kept on ice and stored at −80 °C until processing.

### 2.3. DNA Extraction, Quantitative PCR and 16S rRNA Amplicon Sequencing

Total DNA was extracted from feces and swabs using the DNeasy PowerSoil Kit (Qiagen, Hilden, Germany) with a few modifications to the manufacturer’s protocol, including an extra heating step at 95 °C for 10 min as described before [9]. We added another incubation step after weighing the samples in order to dissolve the fecal and mucosal material from the swab. Feces and swabs were weighed, placed in the bead-beating tube with the lysis buffer, vortexed and incubated for one hour at 4 °C. Bead beating (beat size, 1 mm) was performed using the Speedmill (Analytik Jena, Jena, Germany) for the fecal samples. After the homogenization, feces and swab samples were processed according to the manufacturer’s protocol. The Qubit DNA HS Assay Kit and the Qubit 4 Fluorometer (Thermo Fisher Scientific Inc., Waltham, MA, USA) were used to measure the DNA concentration in the sample eluates. The DNA concentration was adjusted to about 2 ng/µL for the quantitative PCR.

For the quantification of total microbial abundances, established primers for universal bacteria, total archaea, total protozoa and fungi and amplification conditions were used [9] and tested for efficiencies and specificity using melting curve analysis (Appendix A). Absolute quantification of total microbes in fecal samples was performed on a qTOWER real-time PCR system (Analytik Jena GmbH, Jena, Germany) using previously published primer sets (Appendix A). Each 10 μL reaction consisted of 2 ng DNA, 5 μL innuMIX qPCR DSGreen Standard (Analytik Jena GmbH), 400 nM each of forward and reverse primers, and DEPC-treated water (G-Biosciences, St. Louis, MO, USA) in a 96-well plate in duplicate. The amplification comprised an initial denaturation at 95 °C for 5 min, followed by 40 cycles of 95 °C for 10 s, primer annealing at 60 °C for 30 s, and elongation at 72 °C for 30 s, followed by the generation of melting curves with increments of 0.1 °C/s between 55 and 95 °C. Additionally, negative controls were also run on each plate.

Standard curves were prepared from 10-fold serial dilutions (10^7^ to 10^3^ molecules/µL) of the purified and quantified PCR products using pooled DNA from fecal samples of piglets from this study [21]. The final copy numbers were calculated using the following equation: (QM × C × DV)/ (S × V), where QM is the quantitative mean of the copy number, C is the DNA concentration of each sample, DV is the dilution volume of isolated DNA, S is the DNA amount (ng) and V is the weight of the sample (g) subjected to DNA extraction. Amplification efficiencies (E = 10^(−1/slope)^) and coefficients of determination (linearity) can be found in Appendix A.

For the taxonomic composition, the V3-V4 hypervariable regions of the bacterial 16S rRNA gene were amplified using the primers 341F-ill (5′-CCTACGGGNGGCWGCAG-3′) and 802R-ill (5′-GACTACHVGGGTATCTAATCC-3′) using a commercial provider (Microsynth, Balgach, Switzerland). An amplicon of approximately 460 bp was generated for library preparation (Nextera XT DNA Sample Preparation Kit, Illumina, San Diego, CA, USA). PCRs were performed using the KAPA HiFi HotStart PCR Kit (Roche, Baden, Switzerland). Sequencing was performed on an Illumina MiSeq sequencing v2 platform (Illumina) using a paired-end protocol. Obtained reads were demultiplexed, and the adapter sequence from the obtained sequences was removed with cutadapt (https://cutadapt.readthedocs.org/, accessed on 5 June 2023).

### 2.4. Bioinformatical Analysis

Raw sequencing reads (Fastq files) were independently processed, aligned and categorized using the Divisive Amplicon Denoising Algorithm 2 (DADA2) [22], which was run as an R script in R studio (version 1.4.1106). Sequence reads were filtered and chimeras were removed using the “removeBimeraDenovo” command. First, the quality profiles of the forward and reverse reads were evaluated separately. Next, the first 10 nucleotides for each read were trimmed, and the total lengths of reads were truncated to 220 and 200 nucleotides for the forward and reverse reads, respectively, to account for the decrease in the quality score of the further nucleotides. Then, reads containing any ambiguities were removed, as were reads exceeding the probabilistic estimated error of two nucleotides. After dereplication of the filtered data and estimation of error rates, amplicon sequence variants were inferred at 99% identity [22]. The next three steps included the merging of the inferred forward and reverse sequences, the removal of paired sequences that did not perfectly match to control against residual errors and the building of the sequence table. Afterward, chimeras were removed using the removeBimeraDenovo() function, and the taxonomy was assigned using the SILVA 138.1 ribosomal RNA (rRNA) database for bacteria [23]. The resulting taxonomic tables, taxonomic assignment and corresponding metadata were combined to create phyloseq objects within the R phyloseq package (version 1.34.0) [24]. Genera present in <0.05% of samples were discarded prior to analysis. The alpha diversity metrics were calculated using the phyloseq “estimate_richness” function from the rarefied amplicon sequence variant tables. The Chao1 index and the Simpson and Shannon diversity measures were estimated. For beta-diversity, non-metric multidimensional scaling (NMDS), and permutational analysis of variance (PERMANOVA) based on the distance of Bray–Curtis were performed in the vegan R package (version 2.6.4) to assess the significance of differences in bacterial composition [25]. Specifically, non-metric multidimensional scaling (NMDS) ordination plots based on the Bray–Curtis dissimilarity matrix (“metaMDS” function) were used to examine microbiome clustering by age and were visualized using the ggplot2 package [26]. PERMANOVA was performed on the Bray–Curtis dissimilarity matrix using the adonis2 function to evaluate associations among fecal microbial beta-diversity and days of life, nutrition, fecal color and consistency, sample type, and sex. Statistical significance was calculated after 999 random permutations.

Horizontal PLS-DA using the “PLSDA” function was applied to identify the most discriminant bacterial genera using the R package “mixOmics” (version 6.14.0) [27]. To determine the main genera that allowed discrimination of groups for nutrition, fecal consistency and color, and matrix type with the lowest possible error rate in the PLS-DA, we tuned the number of retained variables. Data for bacterial genera were integrated for each day of life. We retained 5% of bacterial genera (relative abundance >0.05%) for components 1 and 2, which comprised 20 taxa. The PLS-DA results (r ≥ 0.4) were visualized as loading plots for the most discriminant variables for each subset of data and component 1.

### 2.5. Statistical Analysis

Differential analysis of the microbiome data was conducted in SAS (version 9.4, SAS Inst. Inc., Cary, NC, USA). The residuals of the microbial gene copy numbers, alpha-diversity indices, and proportional genera abundances (>0.05% abundance in sample) were tested for normal distribution using the Shapiro–Wilk test in SAS. If residuals were not normally distributed, they were converted using the Boxcox method and the Transreg procedure in SAS. Three different repeated models were run to analyze the microbiome data to assess the effects of nutrition, fecal color and consistency and sample type over time. The fixed effects of the first model included day of life, feeding group (sow milk versus additional creep feeding), replicate batch and sex and their two- and three-way interactions. The second model comprised the fixed effects of day of life, fecal color, fecal consistency, replicate batch and sex and the respective two- and three-way interactions. The third model included the fixed effects of day of life, sample type (swab samples versus feces), replicate batch and sex and the respective two- and three-way interactions. A fourth random model was additionally used to analyze the body weight development between feeding groups. Fixed effects included replicate batch, sex, diet and the respective interactions. The random effect was replicate batch and litter. Litter size at birth was considered as a covariate. For the majority of the variables, differences for sex and replicate batch were not observed and were removed from the final models. In all models, the experimental unit was piglet nested within litter. Degrees of freedom were approximated by the Kenward–Roger method. Multiple pairwise comparisons among least-square means were performed using the probability of difference option in SAS. Data were expressed as least-square means ± standard error of the mean (SEM). Significance was defined at *p* ≤ 0.05. For the microbial gene copy numbers, only statistical differences (*p* ≤ 0.05) that were >0.5 log units were discussed as physiologically relevant differences. Descriptive statistics were calculated for creep feed intake during the suckling period using PROC MEANS in SAS.

## 3. Results

### 3.1. Litters and Estimated Creep Feed Intake

The average litter size was 13.1 and 12.5 ± 0.9 (SD) piglets in the sow milk-only and creep feed groups, respectively. The creep feed intake was estimated on litter level. The creep feed intake varied among litters and averaged 10 g (DM basis; milk replacer) per piglet and day between days 10 and 16 of life (Appendix A). Thereafter, its intake increased to 79 g (dry matter basis) per piglet and day when the piglets were transferred to 100% of the prestarter diet in wet form on days 26 to 28 of life. Selected piglets for fecal samplings in the creep feed group showed a lower body weight development than piglets that suckled only sow milk during the suckling and postweaning period (Appendix A).

### 3.2. Temporal Dynamics in Bacterial Microbiome

We evaluated changes in gene copies per gram of wet feces as a measure of absolute microbial abundances for the age-related microbial dynamics in the feces (Table 1, Table 2 and Table 3). Total abundances of bacteria (on average 8.6 log_10_ gene copies/gram of wet feces) were relatively similar over time from day 2 to 34 of life, whereas total archaea increased (*p* < 0.05) by 1.8 log units from day 2 to 20 of life, after which their numbers remained relatively stable until day 34 of life. Protozoa were absent during the suckling phase and only appeared in feces on days 30 and 34 of life. Lastly, total abundances of fungi and yeast were relatively stable from day 2 to 27 of life but increased on days 30 and 34 of life (*p* < 0.05).

For the proportional bacterial composition, an average of 35,339 ± 15,274 amplicon sequence variants were obtained per sample, for a total of 279 fecal samples across days of life, with an increase in species richness (Chao1; *p* < 0.001) and diversity (Shannon, Simpson; *p* ≤ 0.005; Table 1, Table 2 and Table 3) as the piglets aged. Species richness increased throughout the suckling phase from on average 131 on day 2 of life to 337 species (Chao1) on day 27 of life and continued to increase after weaning (418 species on day 30 and 513 species on day of life; *p* < 0.001). Similarly, the diversity increased during the suckling phase from a Shannon index of 2.60 on day 2 to 4.38 on day 27 of life, and again postweaning by 0.1-fold (*p* < 0.05). The PERMANOVA (Bray–Curtis) confirmed that age was an important predictor for differences in the fecal bacterial communities (Bray–Curtis, *p* < 0.001), whereas the sex of pigs did not significantly contribute to the variation in the bacterial community structures (*p* = 0.285). The NMDS (Bray–Curtis distance) indicated five clusters that matched with piglet age (Figure 1). The bacterial communities from the subsequent days of life clustered generally closer together.

The PLS-DA was used to identify the most discriminant genera with increasing age of the piglets (Figure 2). The PLS-DA mainly identified genera for days 2, 6 30 and 34 of life. To provide a few examples, *Escherichia*/*Shigella* were discriminant on day 2 of life, and *Bacteroides* was discriminant on day 6 of life. Most identified bacterial genera, however, were discriminative for day 34 of life, including *Faecalibacterium*, *Blautia*, *Lachnospira* and *Agathobacter* which were less abundant one week postweaning. These results were confirmed by the mixed model analysis (Appendix A). Briefly, the ANOVA showed that on day 2 of life, the fecal microbiome was dominated by *Escherichia/Shigella* (38.1%)*, Fusobacterium* (13.6%)*, Clostridium sensu scrictu* (13.5%) and *Bacteroides* (11.4%). While the dominance of *Escherichia/Shigella* continued (days 6 and 13 of life), *Bacteroides* (day 6 of life) and *Lactobacillus* (days 6 and 13 of life) also increased in their predominance. The three genera declined towards days 20 and 27 of life. Before weaning on day 27 of life, *Bacteroides* (7.4%) and an unclassified genus of the *Rikenellaceae* RC9 gut group (7.0%) dominated the fecal microbiome. Postweaning, *Lactobacillus* dropped to 1.7% (of all reads) on day 30 of life but increased to their preweaning abundance of 5.0% on day 34 of life. While *Prevotella* already increased during the suckling phase to 5.3%, they further increased in their abundance to 6.6 and 7.0% on days 30 and 34 of life, whereas the abundance of *Bacteroides* dropped postweaning to 1.0% on day 34 of life.

### 3.3. Differences in Age-Related Bacterial Development Due to Creep Feeding

Creep feeding decreased total bacterial and fungal abundances by 0.6 and 1.1 log units on day 20 of life, respectively, whereas it increased both groups by 0.6 log units on day 27 of life compared to the other days (Table 1). Protozoal and archaeal abundances were similar between dietary groups. Moreover, the introduction of creep feed from day 10 of life did not alter the age-related development of species richness and alpha-diversity (Shannon, Simpson; *p* > 0.05; Table 1). This was supported by the PERMANOVA, showing little effects of the nutrition during the suckling phase on the bacterial community structure (creep feeding effect, *p* = 0.178; day of life × creep feeding interaction, *p* = 0.069).

Following the age-dependent changes in the predominant taxa, the PLS-DA showed varying discriminant taxa that were influenced by piglet nutrition during the suckling phase for the various days of life (Figure 3). For instance, the PLS-DA identified *Coprococcus* to be the most discriminant for creep feeding, whereas *Sutterella* and *Parabacteroides* were influential in feces of piglets only drinking sow milk on day 13 of life (Figure 3A). On day 20 of life, *Turicibacter* discriminated best in the feces of creep-fed piglets, whereas *Alistipes* was the most influential taxa increased in feces of piglets only drinking sow milk (Figure 3B). The loading plot for day 27 of life indicated that *Actinobacillus* and *Bacteroides* were most depressed due to creep feeding, whereas an unclassified genus (p 1088 a5 gut group) and *Coprococcus* again were raised in feces of piglets only drinking sow milk (Figure 3C). Two days after weaning (day 30 of life), *Prevotella* and *Lachnospiraceae* UCG 004 were the most important and higher abundant taxa in the feces of creep-fed piglets, whereas *Treponema* and *Christensenellaceae* R7 group were most influential in the feces of piglets only drinking sow milk (Figure 3D). One week postweaning (day 34 of life; Figure 3E), *Selenomonas* and *Clamydia* discriminated best in creep-fed piglets, whereas *Foumierella* and an unclassified *Butyriococcaceae* genus (UCG 008) were the most influential taxa in feces of piglets only drinking sow milk. The most influential bacterial genera for the effect of the type of nutrition during the suckling phase that were identified by means of the PLS-DA were also different in the multiple pairwise comparisons of the mixed models for which detailed results for the least-square means and fixed effects can be found in Appendix A.

### 3.4. Differences in Bacterial Abundances between Feces of Varying Color and Consistency

Feces of different colors and consistencies contained different gene copy numbers of bacteria and fungi. Whether dryer feces (balls) contained more or less bacteria and fungi compared to soft and very soft feces depended on the day of life (Table 2). Feces of different colors and consistencies also diverged in their species richness and diversity. From day 20 of life, brown feces were richer in taxa (Chao1, *p* < 0.05; Table 2) and generally more diverse than yellow feces (Shannon; *p* < 0.05). Fecal consistency affected the Simpson index, with soft feces and diarrhea comprising a more diverse community (*p* < 0.05). The PERMANOVA confirmed the impact of fecal color (day of life × fecal color interaction, *p* = 0.002) and consistency (day of life × fecal consistency interaction, *p* = 0.002) on the bacterial community composition on the various ages.

The most discriminating genera for color and consistency varied on the various days of life according to the PLS-DA (Figure 4). As an example, on day 2 of life, the majority of the identified genera discriminated for dark-brown meconium samples, including *Citrobacter*, *Turicibacter*, *Mitsuokella*, *Escherichia/Shigella* and *Megasphara* (Figure 4A). *Lactobacillus*, in turn, was representative for yellow–soft samples, whereas in yellow–ball samples, *Bifidobacterium* was the characteristic taxon on day 2 of life. Brown balls were characterized by *Staphylococcus* on day 2 of life. For day 6 of life (Figure 4B), PLS-DA distinguished influential taxa for brown balls (e.g., *Clostridium* sensu stricto, *Staphylococcus*, *Butyriococcus* and *Megasphaera*) and yellow balls (e.g., *Moraxella* and *Lactobacillus*). Similarly, on day 13 of life (Figure 4C), brown very soft feces were characterized by an unclassified genus of the *Prevotellaceae* NK3B31 group, *Prevotella* and *Alloprevotella*, whereas *Sphaerochaeta*, an unclassified genus within the *Rikenellaceae* RC9 gut group, were discriminative for yellow–soft feces. Yellow balls were discriminated by *Escherichia/Shigella* and *Enterococcus* on day 13 of life. On day 20 of life (Figure 4D), the most discriminative genera were *Monoglobus* for yellow–soft feces, an unclassified *Rikenellaceae* genus from the dgA 11 gut group for grey–soft feces, an unclassified *Ruminococcaceae* genus (Candidatus *Soleaferrea*) for grey balls, *Escherichia/Shigella* for brown balls and an unclassified *Lachnospiraceae* genus for yellow–very soft feces. One day before weaning (day 27 of life, Figure 4E), the most discriminant genera differed again among features of feces. Here, the unclassified genus of the *Prevotellaceae* NK3B31 group and *Prevotella* discriminated for grey–soft feces, whereas *Veillonella* and *Parabacteroides* discriminated for brown very soft feces. *Phascolarctobacterium* and *Escherichia*/*Shigella* were representative for brown soft feces and yellow balls, respectively, whereas brown balls were discriminated by *Blautia*. Postweaning, on day 30 of life (Figure 4F), more discriminative bacteria were identified for brown very soft feces than for the other color–consistency combinations, including *Methanobrevibacter* and *Phascolarctobacterium*. On day 34 of life (Figure 4G), more identified genera were characteristic for brown balls, including *Coprococcus*, an unclassified *Butyriococcaceae* genus (UCG 008), *Blautia* and *Denitrobacterium*. The ANOVA confirmed differences in abundances of bacterial genera in feces of different colors and consistencies. The detailed results can be found in Appendix A.

### 3.5. Differences in Bacterial Abundances between Feces and Swab Samples

The type of sample impacted the total abundances of bacteria (*p* < 0.001) and fungi (*p* = 0.016) on the various days of life (Table 3). Whether feces or swab samples contained more bacteria depended on the day of life. For instance, swab samples comprised 0.6-, 0.7- and 0.7-fold more bacterial gene copy numbers than feces on days 2, 27 and 34 of life, respectively (*p* < 0.05), whereas on day 20 of life, swab samples contained 0.9-fold less bacterial gene copies per gram than feces (*p* < 0.05).

The type of sample did not influence the species richness but affected the bacterial alpha-diversity (Shannon, Simpson; *p* < 0.05). Generally, feces and the combination of feces and swab samples had a higher alpha-diversity than mucosal swab samples across sampling days. For the beta-diversity, the PERMANOVA demonstrated different bacterial community structures in fecal swabs, feces and the combination of swab and fecal samples (day of life × sample type interaction, *p* = 0.013). These observations were supported by the PLS-DA, identifying different discriminative genera for each type of sample (Figure 5). Accordingly, *Citrobacter* was the lead genus in meconium samples, whereas *Bifidobacterium* and *Lactobacillus* were the influential genera in fecal swab samples on day 2 of life (Figure 5A). Feces were discriminated by *Moraxella* and *Rothia*, and the combination of feces and fecal swab samples by *Staphylococcus* on day 2 of life. On day 6 of life (Figure 5B), fecal swab samples were characterized by a lower proportion of *Eggerthellaceae* genus (DNF00809), *Enterococcus* and *Escherichia/Shigella*, whereas feces contained more *Butyricicoccus* and *Rothia*. The combination of both sample types was characterized by more *Bacteroides* and *Intestinimonas* on day 6 of life. The discriminative genera changed again towards day 13 of life (Figure 5C), when the combined samples were characterized by more *Oscillospiraceae* genus (UCG 002), the swab samples by *Terrisporobacter* and *Anaerococcus* and feces by *Intestinimonas* and *Lactobacillus*. On day 20 of life (Figure 5D), swab samples comprised more *Turicibacter*, *Actinomyces* and *Collinsella*, whereas feces were characterized by less *Alistipes*, *Oscillospiraceae* genus (UCG 002) and an unclassified *Rikenellaceae* genus of the RC9 gut group. The most distinguishable bacteria changed again on day 27 of life (Figure 5E), with *Escherichia/Shigella* being discriminative for swab samples, and *Desulfovibrio* and *Phascolarctobacterium* being characteristic for feces and the combination of both, respectively. Postweaning, on day 30 of life (Figure 5F), feces were characterized by less *Megasphaera* and *Phascolarctobacterium*, whereas the combination of feces and swab samples discriminated for *Monoglobus*. Swab samples on day 30 of life were characterized by *Anaerococcus* and *Denitrobacterium*. On day 34 of life (Figure 5G), the influential genera in swab samples were *Blautia* and again *Denitrobacterium*. Feces discriminated for less *Treponema*, and the combination of both sample types discriminated for an unclassified *Ruminococcaceae* genus (Candidatus *Soleaferrea*) on day 34 of life. Similar results were obtained from the mixed model analysis (Appendix A).

## 4. Discussion

For repeated sampling schemes, feces can provide valuable data. When collecting feces, standardization of the collection and sample handling procedures is critical to ensure comparability among studies on the fecal microbiome in piglets [11,21]. However, fecal color, consistency and sample type often receive little attention, although, as the present data show, they are vital to be considered in comprehensive microbiome studies. Here, we provide missing data for the effect of creep feeding on the developing fecal microbiome and variation in the microbiome composition due to differences in fecal characteristics (i.e., fecal color and consistency) and sample type throughout the suckling and early postweaning phase. These data not only add to our knowledge about early-life microbial fluctuation dynamics in neonatal piglets but also provide valuable information for diagnostics and optimization of feed formulations. Our study was designed to consider the effect of gender, which had little influence on the fecal microbiome and growth during the suckling and early postweaning phase. Lastly, in conducting this study under production conditions, the present data are directly transferable to the practice.

Aside from the taxonomic composition, the dynamics in the total microbial load is an important component in the gut microbe–host interplay [8]. The developmental patterns differed among the various microbial groups in the present study. This observation suggested different time windows for the acquisition of bacteria, archaea, protozoa and fungi [8]. Total bacterial, archaeal and fungal abundances were stable throughout the suckling phase from day 2 of life (except archaea which increased on day 13), indicating that their colonization took place within the uterus, during birth or shortly after birth. Also, weaning did not largely modify the measurable bacterial, archaeal and fungal gene copies per gram of feces. However, it can be assumed that the overall microbial load was lower due to the lower feed intake and hence less feces that were excreted. By contrast, the high starch content of the prestarter diet, microbe–microbe interaction or suppressive action of porcine milk compounds may be the reason why protozoa only appeared postweaning, which needs further investigation. Correspondingly, the higher abundance of archaea postweaning may be related to the fiber fraction of the prestarter diet and microbial cross-feeding [28,29]. Sow feces and skin serve as reservoirs of microbes during the suckling period [8]. Therefore, with the creep feed already containing hydrolyzed starch, it may have been thinkable to detect protozoa earlier in the creep-fed piglets. However, the initially low creep feed intake and the high digestibility of the starch component in the milk replacer may have limited the substrate available for protozoa in the large intestine during the suckling phase. Another assumption is that the milk-oriented microbiome and/or milk components suppressed the appearance of protozoa, which also needs further investigation.

The taxonomic composition of the fecal microbiome showed characteristic changes during the suckling and after weaning [6,16,17]. Correspondingly, *Escherichia*, *Clostridium* sensu stricto, *Fusobacteria* and *Bacteroides* were the dominant genera initially (day 3 of life), whereas carbohydrate-fermenting taxa, such as *Lactobacillus* but also *Prevotella,* progressively increased in the present study. Interestingly, *Prevotella* and other unclassified genera within the family *Prevotellaceae*, which were reported to only increase with the onset of eating plant-based feed, started to increase from day 20 and reached their postweaning proportional abundance on day 27 of life. Several reasons may explain this observation, such as the consumption of sow feces or spills from sow feed, or the metabolic flexibility within the family *Prevotellaceae* to utilize various substrates [8]. As illustrated by the Bray–Curtis distances, greater maturational processes occurred from day 3 to 20 of life, whereas the microbiome structure remained similar from day 20 to 27 of life. Weaning did not cause very drastic changes in the fecal microbiome structure on day 30 of life as we would have anticipated from previous research [6,16,17]. Nevertheless, highly abundant and milk glycan-fermenting *Lactobacillus* and *Fusobacterium* dropped on day 30 of life likely due to the lack of porcine milk oligosaccharides [30] and lower feed intake in general. The PLS-DA identified *Roseburia* and an unclassified *Lachnospiraceae* genus as most discriminant on day 30 of life; these are taxa that thrive on plant fiber and starch and are related to butyrate production [31]. A greater abundance of butyrate producers may be beneficial for maintaining the normal function of the enterocytes and controlling mucosal inflammation postweaning [32]. The separate clustering of the microbiomes in the NMDS plots, PLS-DA and PERMANOVA support that a larger reorganization in the fecal microbiome occurred from day 30 to day 34 of life, reflecting the adaptation to the diet without sow milk and increased feed intake. Oppositely, the similar and greater Shannon and Simpson indices on days 30 and 34 compared to day 27 of life supported that the diversification of the microbial community started immediately postweaning irrespective of the potentially lower feed intake on day 30 compared to day 34 of life. The mixing of litters after weaning may have contributed to the diversification of the microbiome on day 30 of life.

The nutrition during the suckling period had a certain influence on the overall bacterial and archaeal load, which may be related to the actual creep feed intake and type. As such, the greater fiber content [28,29] in the prestarter diet may have raised the archaeal numbers on day 27 of life. Although the PLS-DA provided discriminant genera on day 13 of life for both feeding groups, such as *Coprococcus* and *Sutterella*, respectively, they were low-abundance taxa (<1% of all taxa). The milk replacer did not modify the abundances of the major milk glycan fermenters, such as *Lactobacillus*, *Bacteroides* and *Fusobacterium*, and taxa in cross-feeding relationships like *Escherichia* [30] at least at the genus level. This may be related to, on the one hand, the initially low consumption of the milk replacer and, on the other hand, the ingredients of the milk replacer. It is a limitation of this study that we could not measure the individual creep feed intake due to the production setting and only estimated on a litter basis. The creep feed intake varied among litters, potentially being linked to the milk production of the sow and the suckling behavior of the piglets, which probably added to the variation in the gut microbiome among piglets within and across the respective litters. It is worth looking at the composition of the milk replacer, which contained mainly whey powder (43% DM), hydrolyzed wheat starch (29% DM), soy and plant protein (15%), and coconut and palm oil (8% DM). Although the ingredients from plants probably supported different taxa than milk, the whey powder as the main ingredient, even if of bovine origin, may have supported the proliferation of milk glycan-fermenting taxa. Bovine whey permeate is rich in lactose and milk oligosaccharides [33], but the oligosaccharide profile of bovine milk is different from that of porcine milk [30,34]. Nevertheless, the high concentration of lactose and specific oligosaccharide profile in the milk replacer seemed not to have largely affected the present genera abundances. The PLS-DA identified *Coprococcus* and *Turicibacter* as discriminative for creep-fed piglets on days of life 13 and 20, respectively. These genera utilize starch and hemicelluloses [35,36]. Therefore, the plant-based ingredients may have been more influential in the milk replacer than the milk glycans. Of note, on day 27 of life, when the piglets ate a considerable amount of the more-fibrous and plant-based prestarter diet, milk glycan-fermenting *Bacteroides* discriminated for the creep feeding. This finding rejected our hypothesis that plant glycan-degrading *Prevotella* would be discriminative for piglets eating creep feed and the prestarter diet in particular. In fact, *Bacteroides* outnumbered *Prevotella* on day 27 of life. This may be related to the fact that the genus *Bacteroides* can utilize a wide range of substrates [37,38], thriving on the starch and plant protein components of the prestarter diet in the present study. Confirming our hypothesis, the genus *Prevotella* was discriminative on day 30 of life in piglets that received the creep feed. This is an interesting finding as *Prevotella* abundances were similar in piglets from both feeding groups on day 27 of life. Despite similar abundances at genus levels, the species composition was different between feeding groups, potentially comprising more plant glycan- and fiber-degrading species in the creep-fed piglets. Also of note, the postweaning drop in *Bacteroides* [30] was detectable only on day 34 but not on day 30 of life, possibly suggesting that other microbe-to-microbe interactions and not only the presence of milk glycans were behind this observation.

Compositional differences in the bacterial community of feces that vary in color and consistency were mainly reported for the “extreme conditions” in piglets with and without diarrhea [18,39]. Our results provide evidence that feces differing in color and consistency diverge in their total microbial load and taxonomic bacterial composition. We only included 40 piglets in this study; therefore, future studies with a greater sample size and other nutritional interventions should expand the present findings on differences in the microbiome composition in feces of the color and consistency. Total numbers, taxonomic composition and the taxa that discriminated for feces of different color and consistency varied on the consecutive sampling days, corresponding to the age-related dynamics. Fecal color and consistency can be assumed to be indicative of differences in the gut environment and host physiology (e.g., bile production [40]), which may be related to intestinal maturation and milk intake in our study. Future studies should investigate whether fecal color at a specific age can predict the gut developmental stage. There were no consistent patterns for higher or lower numbers of bacteria, archaea, fungi and yeasts, and protozoa in feces of a specific consistency and color. Due to the higher water content, very soft feces may contain a lower microbial load. The present data show that this was not necessarily the case. As an example, the bacterial community in brown–very soft feces on day 27 of life contained more bacteria and archaea than brown balls and a more diverse community than the other classifications of feces. It can be speculated whether the greater diversity resulted in higher short-chain fatty acid production in the distal colon, which may have acted osmotically and explains the higher water content. In accordance with that assumption, these feces were further discriminated by the predominant taxa *Prevotella*, *Alloprevotella* and unclassified *Prevotellaceae* (NK3B31 group), which are very versatile genera in terms of substrate utilization and efficient SCFA producers [41]. Regarding the color, we could collect brown and yellow feces across all days of life. This observation rejects our hypothesis that the fecal color is linked to the nutrition of the piglets and may reflect their creep feed intake. However, as we conducted this study under production conditions, our observation may have been confounded by the fact that piglets play with their mothers’ feces which act as natural microbial inoculum and are a source of non-digested fibrous residues. Second, as mentioned earlier, piglets may have taken up feed spills dripping from the mouth of the sow. By supporting that fecal color and nutrition are not linked, the present results also showed that darker feces did not per se comprise more plant-degrading taxa, e.g., *Prevotellaceae*, compared to yellowish-colored feces. Another interesting finding was that *Escherichia*, as the first gut colonizer, dominated in brown feces and meconium, but this taxon was 0.2-fold less abundant in yellow feces on day 2 of life. This might correspond to the state of gut colonization in piglets excreting brown feces than in piglets with yellow feces. This assumption may be also supported by the Chao1 on day 2 of life, being higher in yellow-colored feces and reflecting a more advanced gut colonization. Future studies should investigate whether darker feces on the first days of life may be used as an indicator for early gut colonization and potentially colostrum and/or milk intake as major microbial substrates. On day 6 of life, brown balls discriminated for butyrate-producing bacteria, including *Clostridium* sensu stricto (butyrate kinase), *Butyricicoccus*, *Megasphaera*, *Christensenellaceae* genus, *Butyricimonas* and *Blautia* (butyryl-CoA-CoA acetate transferase pathway) [42,43,44,45]. The small sample amount did not enable us to determine SCFA in feces to confirm that the piglets with brown balls may benefit from a higher intestinal butyrate production.

With regard to the sample type, Chowdhury et al. [11] concluded previously that the sample type (feces versus swab samples) was not the major source of variation in the fecal microbiome composition in their study. The present results for dissimilarities (PERMANOVA) indicated distinct communities, demonstrating that the comparability between sample types may be limited. The actual effect depended on the day of life and hence on the maturational stage of the microbiome. The three sample types clustered apart with different discriminating bacterial genera that were representative for the mucosal (swab samples) and digesta-associated (feces) bacterial communities. Notably, we expected that the mixed samples (feces + swab samples) would show total and relative abundances that were between feces and swab samples. Results showed that this was not always true and may be linked to the sample collection or sample processing during DNA extraction, which needs to be followed up. Moreover, the ANOVA demonstrated large differences in the abundances of the dominant bacterial genera among sample types on the various days of life, which may interfere with the interpretation of age- and nutrition-related dynamics in the fecal microbiome. Our data also demonstrate the importance of analyzing meconium and fecal samples separately on the first days of life. To give an example, the proportion of *Escherichia* was 1.4-fold less in feces compared to the meconium, clearly increasing the inter-animal variation if sample types were combined.

Lastly, it should be mentioned that the present study was not designed to assess the milk and feed intake of the piglets. However, the feed intake, not only postweaning but also the milk intake during the suckling period, may affect the outcome for the fecal microbiome due to changes in passage rate and the available substrate along the gastrointestinal tract, which should be considered in future studies. In fact, differences in the suckling behavior of the selected piglets may explain creep-fed piglets gaining less weight compared to the piglets that only suckled sow milk. Other piglets from these litters grew similarly during the suckling phase and only showed a difference in body weight between the feeding groups postweaning on day 34 of life [19], which may be related to a lower postweaning feed intake in the creep-fed piglets.

## 5. Conclusions

The present results demonstrate that age and nutrition during the suckling phase were major factors influencing total and relative microbial abundances. Moreover, our findings show different taxonomic compositions in feces of different consistency, color, and sample type. These findings indicate that these characteristics should be considered in future microbiome studies using feces. The information about the age- and nutrition-related fluctuation in the microbiome is useful as it adds to our knowledge about early-life microbial dynamics, which is valuable for optimizing feed formulations for neonatal piglets. The information that we gained about the differences in microbiome composition in feces of differing in color, consistency and sample type may be helpful in diagnostics. Due to the limited sample size, a higher number of animals and other nutritional interventions should be included in future studies to expand the present findings on the effect of the color, consistency and sample type on the development of the fecal microbiome composition and potential consequences for the host.

## Figures and Tables

**Figure 1 animals-13-02251-f001:**
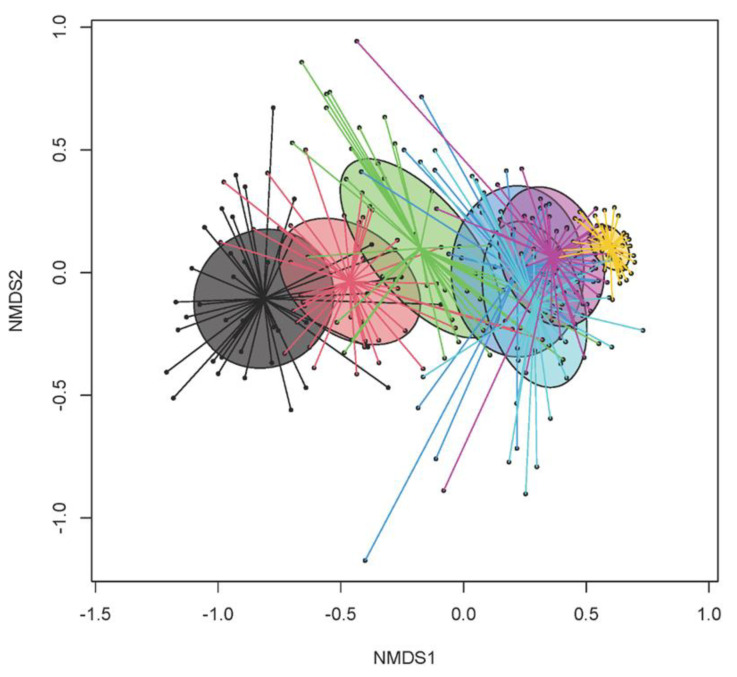
Non-metric multidimensional scaling (NMDS) plot of pairwise Bray–Curtis dissimilarities between bacterial communities in feces of suckling and newly weaned piglets across nutritional groups, features of feces and matrices (stress level = 0.19). Ellipses represent the standard deviation. Days 2 (black), 6 (red), 13 (green), 20 (dark blue), 27 (blue), 30 (purple) and 34 (yellow) of life. Weaning took place on day 28 of life.

**Figure 2 animals-13-02251-f002:**
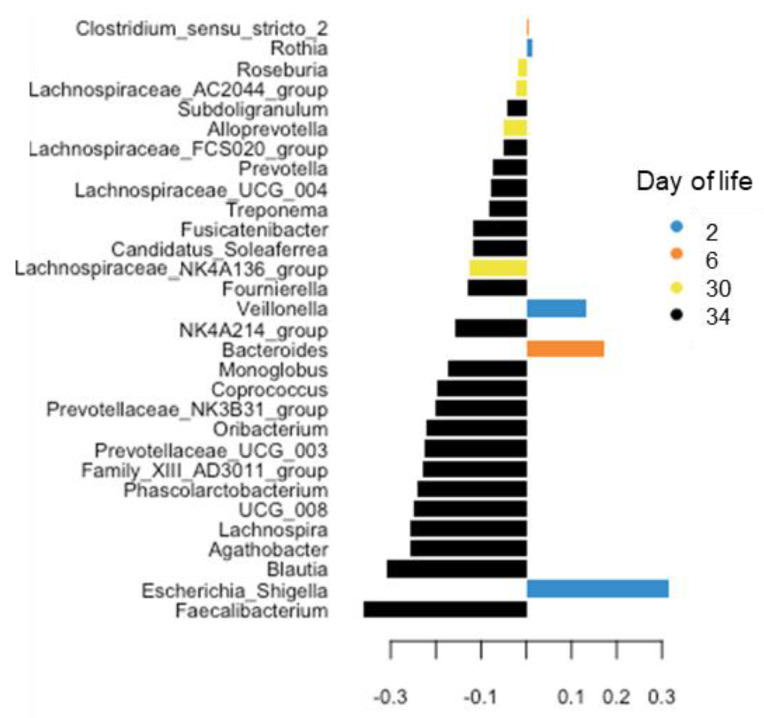
Loading plots of sparse partial square discriminant analysis displaying the most discriminant genera for age in feces of piglets during the suckling and early postweaning period. Weaning took place on day 28 of life. Discriminant bacterial genera were identified for days 2, 6, 30 and 34 of life but not for days 13, 20 and 27 of life.

**Figure 3 animals-13-02251-f003:**
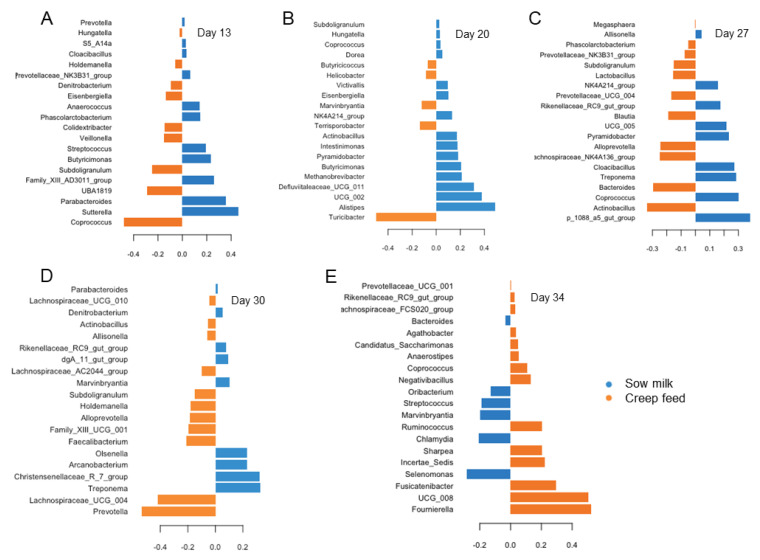
Loading plots of sparse partial square discriminant analysis displaying the most discriminant genera for the feeding in the suckling phase in feces of piglets receiving only sow milk or additionally creep feed (blue) from day of life 10 (orange) at day of life (**A**) 13, (**B**) 20, (**C**) 27, (**D**) 30 and (**E**) 34.

**Figure 4 animals-13-02251-f004:**
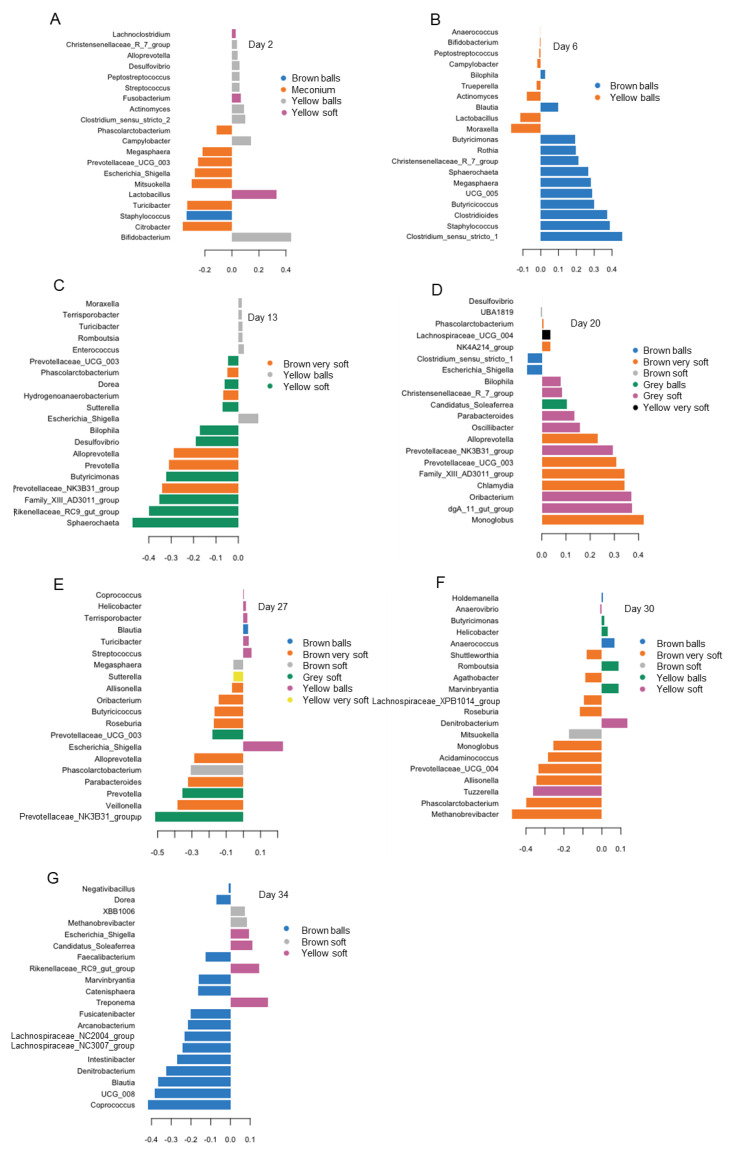
Loading plots of sparse partial square discriminant analysis displaying the most discriminant genera for color and consistency of feces in piglets at day of life (**A**) 2, (**B**) 6, (**C**) 13, (**D**) 20, (**E**) 27, (**F**) 30 and (**G**) 34. Color types: yellow, gray and brown. Consistency types: balls, soft feces and very soft feces. Meconium characteristics: dark brown color and soft consistency.

**Figure 5 animals-13-02251-f005:**
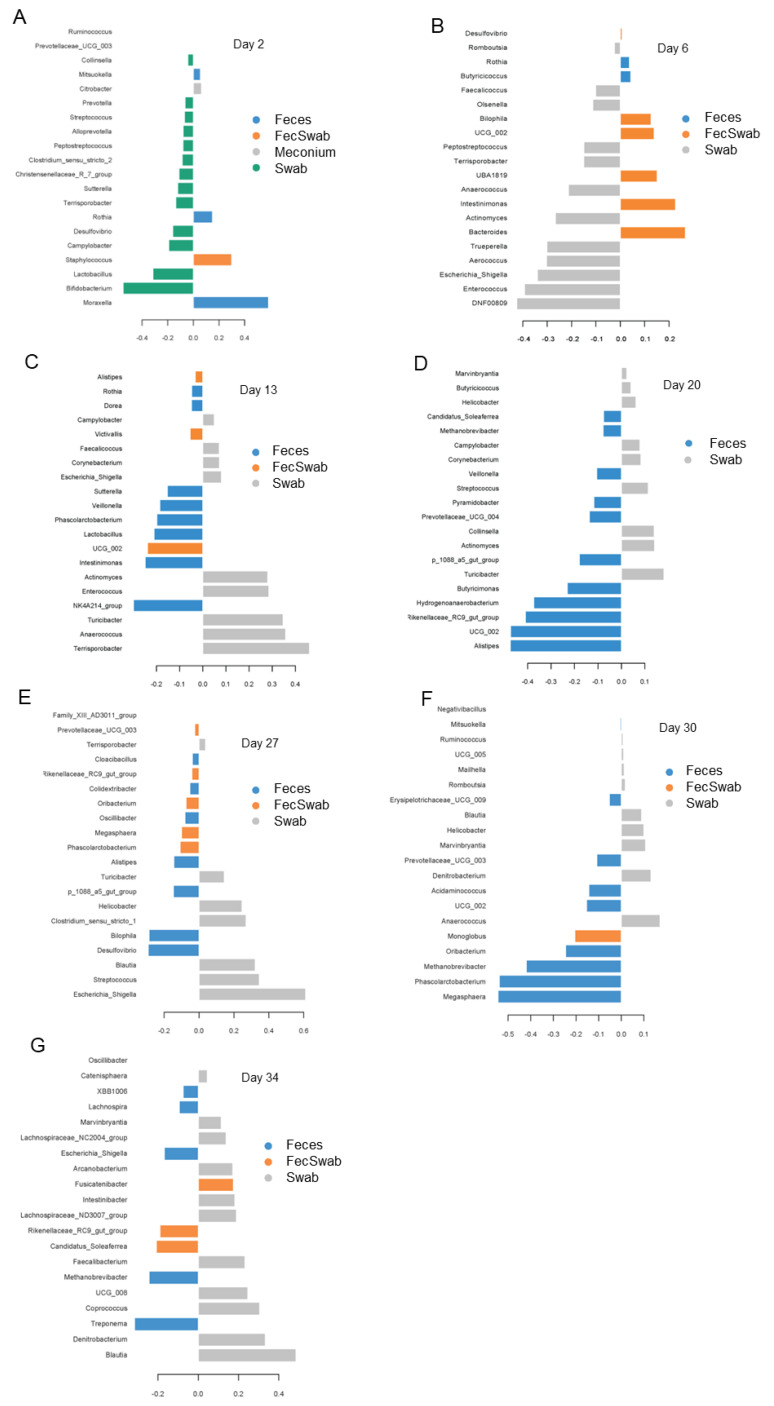
Loading plots of sparse partial square discriminant analysis displaying the most discriminant genera for fecal sample type in piglets at day of life (**A**) 2, (**B**) 6, (**C**) 13, (**D**) 20, (**E**) 27, (**F**) 30 and (**G**) 34. Sample types: feces, swabs, the combination of feces and swabs (FecSwab), meconium.

**Table 1 animals-13-02251-t001:** Age-related development of total microbial abundances (log_10_ gene copies/g sample), species richness and alpha-diversity indices in feces from suckling and newly weaned piglets fed either only sow milk or additional creep feed from day 10 of life *.

Day of Life (DoL)	2	6	13		20		27		30		34			*p*-Value		
Feeding (Feed)	Sow Milk	Sow Milk	Sow Milk	Creep Feed	Sow Milk	Creep Feed	Sow Milk	Creep Feed	Sow Milk	Creep Feed	Sow Milk	Creep Feed	Pooled SEM	DoL	Feed	DoL × Feed
Total bacteria	8.5	8.6	8.4	8.6	8.8	8.3	8.1	8.7	8.5	8.4	8.7	8.8	0.17	0.673	0.785	0.022
Total archaea	3.6	3.9	4.9	5.1	5.9	4.8	4.6	5.2	5.6	5.1	5.6	5.9	0.22	<0.001	0.313	0.001
Fungi and yeasts	3.8	3.9	3.8	4.0	4.0	4.2	3.8	3.9	4.5	4.4	4.2	4.6	0.17	<0.001	0.182	0.855
Total protozoa	ND	ND	3.6	ND	4.5	ND	ND	4.6	4.4	4.2	5.8	5.5	0.44	<0.001	0.978	0.918
Chao1	133	189	227	200	309	294	308	304	448	450	539	588	26.61	<0.001	0.898	0.897
Shannon	2.62	3.33	3.48	3.38	4.20	3.87	4.13	4.15	4.71	4.62	4.98	5.24	0.14	<0.001	0.332	0.258
Simpson	0.80	0.89	0.90	0.88	0.96	0.92	0.96	0.96	0.97	0.97	0.98	0.99	0.02	<0.001	0.220	0.501

* Values are least-square means ± pooled standard error of the mean (SEM). Piglets were weaned on day 28 of life. ND, not detected.

**Table 2 animals-13-02251-t002:** Differences in total microbial abundances (log_10_ gene copies/g sample), species richness and alpha-diversity indices in feces of different colors and consistencies obtained from suckling and newly weaned piglets during the suckling and early postweaning phase *.

**Day of Life (DoL)**	**2**	**2**	**2**	**2**	**2**	**6**	**6**	**6**	**13**	**13**	**13**	**13**	**20**	**20**	**20**	**20**	**20**	**20**	**20**	**20**
**Color**	**Brown**	**Brown**	**Yellow**	**Yellow**	**Yellow**	**Brown**	**Yellow**	**Yellow**	**Brown**	**Brown**	**Yellow**	**Yellow**	**Brown**	**Brown**	**Brown**	**Grey**	**Grey**	**Yellow**	**Yellow**	**Yellow**
**Consistency**	**Balls**	**Meconium**	**Balls**	**Very Soft**	**Soft**	**Balls**	**Balls**	**Soft**	**Balls**	**Very Soft**	**Balls**	**Soft**	**Balls**	**Very Soft**	**Soft**	**Balls**	**Soft**	**Balls**	**Very Soft**	**Soft**
Total bacteria	8.3	8.1	8.7	8.5	8.6	8.7	8.6	8.4	8.2	8.8	8.6	9.0	8.7	8.4	9.3	9.2	9.0	8.4	8.7	8.9
Total archaea	2.6	3.5	3.8	4.3	2.6	3.4	3.9	4.0	4.6	5.1	5.1	5.1	6.0	6.0	6.3	5.4	6.2	5.2	3.9	5.9
Fungi and yeasts	3.3	3.5	4.0	4.4	3.4	3.5	4.0	4.6	3.9	3.4	3.9	3.6	4.1	4.8	4.2	4.6	4.6	4.1	3.7	4.3
Total protozoa	ND	ND	ND	ND	ND	ND	ND	ND	ND	ND	ND	ND	ND	ND	ND	ND	ND	ND	ND	ND
Chao1	112	101	143	147	151	189	189	176	216	185	212	250	274	540	233	471	437	292	226	346
Shannon	2.48	2.10	2.75	2.60	3.07	3.39	3.31	3.60	3.51	3.70	3.36	4.14	3.84	5.02	4.18	4.71	4.83	3.96	3.90	4.34
Simpson	0.83	0.72	0.81	0.83	0.90	0.91	0.89	0.95	0.90	0.94	0.88	0.95	0.94	0.97	0.96	0.96	0.96	0.93	0.96	0.96
**Day**	**27**	**27**	**27**	**27**	**27**	**27**	**27**	**30**	**30**	**30**	**30**	**30**	**34**	**34**	**34**	**34**	**34**			
**Color**	**Brown**	**Brown**	**Brown**	**Grey**	**Yellow**	**Yellow**	**Yellow**	**Brown**	**Brown**	**Brown**	**Yellow**	**Yellow**	**Brown**	**Brown**	**Brown**	**Yellow**	**Yellow**			
**Consistency**	**Balls**	**Very Soft**	**Soft**	**Soft**	**Balls**	**Very Soft**	**Soft**	**Balls**	**Very Soft**	**Soft**	**Balls**	**Soft**	**Balls**	**Very Soft**	**Soft**	**Balls**	**Soft**	**Pooled SEM**		
Total bacteria	7.4	8.7	8.9	9.0	8.8	7.4	8.6	8.2	8.4	9.2	8.7	9.5	9.1	8.5	8.5	9.3	8.4	0.42		
Total archaea	3.8	4.7	5.6	5.6	5.4	3.0	5.4	5.0	5.6	6.0	5.4	6.0	5.9	5.5	5.8	5.2	6.1	0.52		
Fungi and yeasts	4.2	3.4	4.2	3.6	4.2	2.3	3.6	4.7	4.5	3.7	4.4	4.3	4.7	3.7	5.0	5.7	3.1	0.38		
Total protozoa	ND	ND	ND	ND	ND	ND	ND	3.8	5.1	5.1	4.4	ND	5.8	5.6	5.6	4.9	6.8	0.71		
Chao1	274	518	317	389	290	327	242	479	448	493	378	694	596	523	597	583	268	65.91		
Shannon	4.05	4.78	4.54	4.50	4.04	4.24	3.85	4.80	4.63	5.20	4.32	5.37	5.34	4.90	5.17	4.26	4.63	0.36		
Simpson	0.95	0.96	0.99	0.96	0.96	0.95	0.94	0.98	0.95	1.01	0.96	0.98	1.00	0.98	0.97	0.95	0.97	0.04		
	***p*-Value**																			
	**DoL**		**Color**		**DoL × Color**		**Consistency**		**DoL × Consistency**		**Color × Consistency**		**DoL × Color × Consistency**							
Total bacteria	0.857		0.764		0.908		0.324		0.638		0.332		0.037							
Fungi and yeasts	0.042		0.236		0.718		0.077		<0.001		0.056		0.041							
Total protozoa	0.034		0.428		0.339		0.214		0.244		0.348		0.421							
Total archaea	<0.001		0.474		0.558		0.163		0.055		0.029		0.138							
Chao1	<0.001		0.005		0.646		0.928		0.004		0.026		0.038							
Shannon	<0.001		0.043		0.756		0.342		0.554		0.499		0.462							
Simpson	0.005		0.684		1.000		0.027		0.979		0.993		0.951							

* Values are least-square means ± pooled standard error of the mean (SEM). Piglets were weaned on day 28 of life.

**Table 3 animals-13-02251-t003:** Differences in total microbial abundances (log_10_ gene copies/g sample), species richness and alpha-diversity indices in different fecal sample types obtained from suckling and newly weaned piglets during the suckling and early postweaning phase *.

Day of Life (DoL)	2				6			13			20			27			30			34				*p*-Value		
Sample Type	F + S	F	M	S	F + S	F	S	F + S	F	S	F + S	F	S	F + S	F	S	F + S	F	S	F + S	F	S	Pooled SEM	DoL	Type	DoL× Type
Total bacteria	8.4	8.2	8.1	8.8	8.8	8.6	8.4	8.8	8.5	8.4	8.6	8.9	8.0	8.8	8.1	8.8	8.7	8.1	8.6	8.7	8.4	9.1	0.25	0.887	0.115	<0.001
Total archaea	2.6	4.3	3.6	3.9	3.5	3.6	4.0	5.1	4.6	5.1	5.7	5.7	4.8	5.1	4.5	5.4	5.4	5.3	5.3	5.5	5.7	5.8	0.36	<0.001	0.775	0.012
Fungi and yeasts	3.1	4.9	3.3	4.1	3.4	3.8	4.3	3.8	3.5	4.2	3.6	4.1	4.3	3.6	3.6	4.2	4.3	3.9	4.7	5.3	4.0	4.7	0.24	<0.001	<0.001	0.016
Total protozoa	ND	ND	ND	ND	ND	ND	ND	ND	ND	ND	ND	ND	ND	ND	ND	ND	4.8	5.2	3.9	4.5	5.8	5.8	0.44	0.053	0.119	0.049
Chao1	101	181	93	155	167	171	210	197	218	217	259	300	317	337	327	268	478	371	479	623	525	595	40.64	<0.001	0.466	0.303
Shannon	2.64	3.18	2.08	2.75	3.40	3.37	3.28	3.76	3.64	3.12	3.97	4.20	3.85	4.39	4.27	3.88	4.90	4.39	4.74	5.45	4.97	5.20	0.22	<0.001	0.031	0.148
Simpson	0.86	0.90	0.72	0.80	0.92	0.91	0.87	0.94	0.91	0.86	0.95	0.96	0.91	0.97	0.96	0.95	0.98	0.96	0.97	0.99	0.98	0.99	0.03	<0.001	0.001	0.681

* Values are least-square means ± pooled standard error of the mean (SEM). Piglets were weaned on day 28 of life. Sample type: F, feces; M, meconium; S, swab; F + S, combined feces + swab sample.

## Data Availability

Raw sequence data can be found at NCBI (PRJNA980418).

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
