# Peer review of "Temporal Microbial Dynamics in Feces Discriminate by Nutrition, Fecal Color, Consistency and Sample Type in Suckling and Newly Weaned Piglets"

_animals, 2023, doi:10.3390/ani13142251_

Round 1

Reviewer 1 Report

Temporal Microbial Dynamics in Feces Discriminate by Nutrition, Fecal Color, consistency and Sample Type in Suckling and Early Weaned Piglets

The manuscript addresses a topic of interest, and it is presented in a sound manner. Methodology is adequate.

Comment. A number of facts (1.- age-related alterations in microbiome, 2. nutrition, 3. fecal consistency, 4. fecal color, 5. sampling system sample) are studied and presented altogether which makes difficult to follow and to extract clear conclusion.

Statistics. Please, indicate in Mat and meth section what was the experimental unit. Was it the piglt, which was replicated over the time?

Figure 3 Legend for D27 is missing

Table 1. Did you use a repeated measurement test? If so, both SEM of main effects and interactions, and SEM of the mean of time and interaction should be included in Tables.

Table S1. Diets are a little confusing. I suggest including a different footnote for each diet. FND concentration in prestárter diet (15.2%) is very high, and theoretical concentration of hemicellulose (aprox 10% is also high). Is there any reason for this?  I suppose this may affect microbion¡me composition and fermentative pattern. Did you include enzyme supplementation?

Lactose concentration is not identified in the diets. I suggest including this concentration in tables, as it may markedly affect lactobacillacea concentration. Apparently Lactose concentration is very high in creep diet (could it be close to 20%?, This is a very high concentration of lactose). I wonder if this sis in concordance to Figure 3C results. A comment on possible relationship of dieaty peculiarities and microbiome results would be welcomed in discussion.     

3.1 (Ln 301) Did you get any evidence of piglets which actually ate creep feeding? I wonder if differences in microbiome composition between two piglets samples from the same sow might provide evidence of piglets which actually ate (eaters) from those which did not (non-eaters)

Author Response

Temporal Microbial Dynamics in Feces Discriminate by Nutrition, Fecal Color, consistency and Sample Type in Suckling and Early Weaned Piglets. The manuscript addresses a topic of interest, and it is presented in a sound manner. Methodology is adequate.

AUTHORS: Thank you very much for your valuable comments.

Comment. A number of facts (1.- age-related alterations in microbiome, 2. nutrition, 3. fecal consistency, 4. fecal color, 5. sampling system sample) are studied and presented altogether which makes difficult to follow and to extract clear conclusion.

AUTHORS: Indeed, several factors were investigated together. We modified our conclusion (L 692-697).

Statistics. Please, indicate in Mat and meth section what was the experimental unit. Was it the piglt, which was replicated over the time?

AUTHORS: The experimental unit was provided (already in the originally submitted manuscript): ‘piglet within litter’ (L 294-295).

Figure 3 Legend for D27 is missing

AUTHORS: Apologies. We added the legend.

Table 1. Did you use a repeated measurement test? If so, both SEM of main effects and interactions, and SEM of the mean of time and interaction should be included in Tables.

AUTHORS: Thank you for this comment. However, we prefer presenting the pooled SEM for reasons of readability and clarity. However, we indicated that pooled SEMs are presented in Tables.

Table S1. Diets are a little confusing. I suggest including a different footnote for each diet. FND concentration in prestárter diet (15.2%) is very high, and theoretical concentration of hemicellulose (aprox 10% is also high). Is there any reason for this?  I suppose this may affect microbion¡me composition and fermentative pattern. Did you include enzyme supplementation?

AUTHORS: Different footnotes for each diet were provided. Regarding the NDF content of the prestarter diet, the main ingredients were oat flakes and barley which are rich in NDF or hemicelluloses (e.g. beta-glucans). Therefore, analyzed NDF proportion is in the acceptable range. The supplemented enzymes were listed under ‘Technological additives: 1,000 IU of phytase, 1,500 EPU of xylanase, 11 mg of beta hydroxy acid, 21 mg of butylated hydroxytoluene, 11 mg of propyl gallate’.

Lactose concentration is not identified in the diets. I suggest including this concentration in tables, as it may markedly affect lactobacillacea concentration. Apparently Lactose concentration is very high in creep diet (could it be close to 20%?, This is a very high concentration of lactose). I wonder if this sis in concordance to Figure 3C results. A comment on possible relationship of dieaty peculiarities and microbiome results would be welcomed in discussion.     

AUTHORS: We did not specifically analyze the lactose concentration in the diets. However, with whey powder being the main ingredient in the milk replacer, the lactose content was probably around 20%. Due to the fact that we analyzed the microbiota composition in feces, the question arises how much lactose reached the distal large intestine. Most of the lactose was probably fermented in the stomach, small intestine and cecum. In addition the host probably also digested a large part of the lactose in the small intestine. In anyway, we modified the paragraph in which we discussed the ingredients of the creep feed (L 590-597).

3.1 (Ln 301) Did you get any evidence of piglets which actually ate creep feeding? I wonder if differences in microbiome composition between two piglets samples from the same sow might provide evidence of piglets which actually ate (eaters) from those which did not (non-eaters)

AUTHORS: Thank you for this comment. We observed the litters to get an idea if piglets consumed creep feed. From these observations, most of the piglets had their snout in the trough. Nevertheless, it is difficult to say if the piglets drank the milk replacer or only played with it. It may be a good idea to evaluate microbiome differences between siblings that drank milk replacer or not. Due to the fact that animal-to-animal differences exist in the microbiome development within a litter, the experimental set-up would need to be different. Probably, the piglets receiving the milk replacer should be force-fed to ensure consumption of a certain amount of the milk replacer to get an idea about the animal-to-animal variation.

Reviewer 2 Report

The authors report and discuss their findings related to fecal microbiota in piglets during suckling period and a few days post-weaning, and its relationship with feed type. The effect of sample type on fecal microbiota has also been investigated.  This is a very interesting manuscript and will add to the existing knowledge in this field.

However, the relationship between microbiome and fecal consistency/color cannot be analyzed without inclusion of feed type. Feed is a determinant of the gut microbiome composition.  Feed may also determine the consistency and color of feces directly (ex. presence of undigested feed ingredients) or indirectly through altering the gut microbiome and metabolomes.  I would suggest addressing this in discussion section. revising the section 3.4 in the results section and any related discussion in the discussion sections accordingly.  For instance, in line 510-511, I don’t think fecal microbiome is developing based on fecal consistency/color. Or in 608-609, l687-689: I don’t think fecal consistency/color has effect on microbiome. In fact, microbiome may affect the fecal consistency/color. Please reword.

Also, I’m not sure why three separate models were run to evaluate the association of microbiome with the independent variables? I would use one model with all independent variables included (if significant). This will allow to determine any possible relationship between feeding type and fecal consistency/color and sample type.

Also how has the “sow” or “litter” effect been treated in the analysis? I would include it as random effect in microbiome model (like it was done in growth performance model).

L145: How about sow parity? The older sows may have different vaginal and fecal microbiome compared to younger sows and gilts.

L341-342, and Figure 2: Why days of 13, 20, and 27 were excluded from PLS-DA?

L560-566: Mixing piglets with others may be another reason for higher diversity in fecal microbiome post-weaning.

L623: SCFA? Short-chain fatty acids?

Author Response

Comments and Suggestions for Authors

The authors report and discuss their findings related to fecal microbiota in piglets during suckling period and a few days post-weaning, and its relationship with feed type. The effect of sample type on fecal microbiota has also been investigated.  This is a very interesting manuscript and will add to the existing knowledge in this field.

AUTHORS: Thank you very much for your valuable comments.

However, the relationship between microbiome and fecal consistency/color cannot be analyzed without inclusion of feed type. Feed is a determinant of the gut microbiome composition.  Feed may also determine the consistency and color of feces directly (ex. presence of undigested feed ingredients) or indirectly through altering the gut microbiome and metabolomes.  I would suggest addressing this in discussion section. revising the section 3.4 in the results section and any related discussion in the discussion sections accordingly.  For instance, in line 510-511, I don’t think fecal microbiome is developing based on fecal consistency/color. Or in 608-609, l687-689: I don’t think fecal consistency/color has effect on microbiome. In fact, microbiome may affect the fecal consistency/color. Please reword.

AUTHORS: Thank your for these points. We totally agree that the feed is one of the major factors shaping the gut microbiome composition and that the microbial composition influences fecal color/consistency. Of course, the present microbiomes were influenced by the creep feeding. However, there must be a misunderstanding. We did not analyze the “effect of color and consistency on the microbiome composition” but how the microbiomes differ in feces of different color and consistency. We checked the discussion for clarity.

Also, I’m not sure why three separate models were run to evaluate the association of microbiome with the independent variables? I would use one model with all independent variables included (if significant). This will allow to determine any possible relationship between feeding type and fecal consistency/color and sample type.

AUTHORS: Thank you for this suggestion. There are certainly different options to test for differences in the microbiome composition. At the beginning, we also considered to analyze all independent variables together in one model. However, our major aim was to investigate whether total and relative microbial abundances would differ in feces of different colors and consistencies per se. If there is a difference, this difference should be detectable across nutritional regimes. The same is true for the sample type. Interactions with the nutrition of the piglet are interesting but should be looked up in detail in a further study. In order to do, the sample size should be larger.

Also how has the “sow” or “litter” effect been treated in the analysis? I would include it as random effect in microbiome model (like it was done in growth performance model).

AUTHORS: Thank you. The microbiome data were analyzed as repeated measures. Litter was considered as already provided in the originally submitted manuscript: “In all models, the experimental unit was piglet nested within litter.” (L 294-295).

L145: How about sow parity? The older sows may have different vaginal and fecal microbiome compared to younger sows and gilts.

AUTHORS:  Sows were of parities 1 to 6 and balanced for the nutritional groups (L 142). We added this to the Materials and Methods section (L 163-164). However, this study can be seen as a pilot study. Follow-up studies are certainly warranted investigating potential influencing factors. as we mentioned in the Discussion section.

L341-342, and Figure 2: Why days of 13, 20, and 27 were excluded from PLS-DA?

AUTHORS: These days were not excluded. The PLS-DA did not provide discriminant genera for these days of life. We added this statement to the legend.

L560-566: Mixing piglets with others may be another reason for higher diversity in fecal microbiome post-weaning.

AUTHORS: Thank you. This could be true and was added to the Discussion (L 569-570).

L623: SCFA? Short-chain fatty acids?

AUTHORS: Sorry. SCFA was written out (L 632).